# A Hybrid Model for Forecasting Realized Volatility Based on Heterogeneous Autoregressive Model and Support Vector Regression

Yue Zhuo [1] and Takayuki Morimoto [2],*

1   Graduate School of Science and Technology, Kwansei Gakuin University, 1 Gakuen Uegahara, Sanda 669-1330, Hyogo, Japan; yue-zhuo@kwansei.ac.jp
2   School of Science, Kwansei Gakuin University, 1 Gakuen Uegahara, Sanda 669-1330, Hyogo, Japan
*   Correspondence: morimot@kwansei.ac.jp

**Abstract:** In this study, we proposed two types of hybrid models based on the heterogeneous autoregressive (HAR) model and support vector regression (SVR) model to forecast realized volatility (RV). The first model is a residual-type model, where the RV is first predicted using the HAR model, and the residuals are used to train the SVR model. The residual component is then predicted using the SVR model, and the results from both the HAR and SVR models are combined to obtain the final prediction. The second model is a weight-based model, which is a combination of the HAR and SVR models and uses the same independent variables and dependent variables as the HAR model; we adjust the contribution of the two models to the predicted values by giving different weights to each model. In particular, four volatility models are used in RV forecasting as basic models. For empirical analysis, the RV of returns of the Tokyo stock price index and five individual stocks of TOPIX 30 is used as the dataset. The empirical results reveal that according to the model confidence set test, the weight-type model outperforms the HAR model and the residual-type HAR–SVR model.

**Keywords:** forecasting; realized volatility; heterogeneous autoregressive model; support vector regression; TOPIX 30





## 1. Introduction

Volatility plays an important role in risk management. However, there is an inherent problem with volatility: real volatility is latent and is not directly observable. Volatility can be estimated using other approaches, such as the autoregressive conditional heteroskedasticity (ARCH) and generalized autoregressive conditional heteroskedasticity (GARCH) models proposed by Engle (1982) and Bollerslev (1986). Such models have been developed and studied extensively; however, because GARCH models use open-to-close return data, they are still subject to unobserved latent volatility. To overcome this problem and take full advantage of information from high-frequency intraday data, Andersen et al. (2003) proposed a framework to compute the summation of the squared high-frequency intraday returns to construct realized volatility (RV) using high-frequency data for measuring, modeling, and forecasting latent volatility. When the measurement error of the RV is ignored, volatility essentially becomes "observable".

For forecasting RV, Corsi (2009) introduced a simple and easy-to-implement model called the heterogeneous autoregressive (HAR) model. This model utilizes the past daily, weekly, and monthly RVs called 1, 5, and 22 lags, respectively, as variables to predict the future daily RV. The HAR model captures the long-term memory and persistence of essential features in financial data. The goal of (G)ARCH models is to estimate and forecast unobservable latent volatility, whereas RV is a nonparametric estimator of latent volatility, and autoregressive moving average (ARMA)-type models, including the HAR model, are used

to predict observable estimators, which are distinguished from each other by the difference in the dependent variables Guidolin and Pedio (2018), McAleer and Medeiros (2008).

Since its introduction, the HAR model has been widely used and continuously refined. Corsi and Renò (2012) further developed the HAR model into the HAR with component jumps (HAR-CJ) model by decomposing RV into continuous and jump components. Building on the realized semivariance, Patton and Sheppard (2015) proposed a HAR-semivariance (HAR-SV) model and demonstrated that it is better than the HAR-CJ model. Bollerslev et al. (2016) considered the distribution of measurement error and realized quarticity (RQ) as an estimator of the variance in the measurement error to construct the HARQ model, which outperformed the other HAR series models in an empirical analysis. Wen et al. (2016) used alternative risk measures to construct a series of HAR-type models for predicting volatility in crude oil futures. Audrino et al. (2018) proposed the flexible HAR (1, 2, ..., *p*) model, which goes beyond the traditional HAR (1, 5, 22) model. This model treats all past p days of RV as estimators, using the least absolute shrinkage and selection operator (LASSO) method with a regularization term to estimate the model, which mitigates the problem of overfitting. Lyócsa and Stašek (2021) introduced a method to improve the predictive accuracy of the HAR model by combining multiple results from HAR models with different volatility estimators. This approach involves averaging the predictions of all the models by using different volatility estimators to derive the final prediction.

In recent years, several studies have used machine learning methods directly in financial forecasting or implementing time series models to improve forecasting accuracy. Various studies by Gupta et al. (2023), Ramos-Pérez et al. (2019), Demirer et al. (2020), Carr et al. (2019), and others have discussed the prediction of RV in financial markets using individual machine learning models such as artificial neural networks and support vector machine (SVM).

In addition to this, studies have combined machine learning models with financial time series models to better capture both linear and nonlinear patterns in time series. Kim and Won (2018) combined long short-term memory (LSTM)-based methods with a GARCH-type model to construct numerous hybrid models. The empirical results with returns data of the KOSPI 200 index revealed that the LSTM-based hybrid models are better than single GARCH-type models in prediction accuracy. Zhang and Qiao (2021) and Sun and Yu (2020) proposed HAR- and GARCH-type support vector regression (SVR) models, respectively. Their finding showed that SVR effectively improved the prediction accuracy of HAR-type and GARCH-type models. Pai and Lin (2005), Li et al. (2010), and Zhu et al. (2017) investigated a hybrid model based on the autoregressive integrated moving average (ARIMA) and SVR or SVM for time series forecasting in different fields, revealing that a hybrid model can improve on the ARIMA model in prediction accuracy.

However, in these studies, the researches did not consider the optimal hyperparameter setting of machine learning models when combining machine learning with financial time series models. Therefore, in this study, following Huang (2012), we use the genetic algorithm (GA) method to optimize the hyperparameters of the SVR model so that we can choose more appropriate hyperparameters. Based on this automated machine learning model, we propose two types of hybrid models that combine the HAR model with the SVR to predict the RV of stock indices and individual stocks returns. The first type predicts the residual of the HAR model predictions using the GA-optimized SVR model as a nonlinear component. Then, we use the sum of the two predicted components as the predicted value of volatility. The second is a weight-type model, where the predicted value $\hat{y}$ is described as $\alpha\hat{y}_{HAR} + (1-\alpha)\hat{y}_{SVR}$, which captures the nonlinear component. We construct the two types hybrid models based on four basic HAR-type models. Then, we compare the performance of these models in out-of-sample forecasting. Additionally, we use other extensions of the HAR model, such as HAR-semivariance (HAR-SV), HAR-signed jumps (HAR-SJ), and HAR-RQ (HARQ), in this study and combine them with SVR. In the empirical analysis, we collect high-frequency intraday price data (and, thus, computing the returns) from the

Tokyo stock price index (TOPIX) and five individual stocks of TOPIX 30 in the Japanese market from 2020 to 2022 as our dataset for an out-of-sample forecasting test to compare the performance of the hybrid models and basic HAR-type model. Our main contributions are as follows: (1) we propose that the problem of parameter tuning in RV prediction using machine learning models can be made more efficient using automatic machine learning models; (2) we apply the optimization algorithm not only for hyperparameter optimization but also in the selection of weights when combining different predictive values; and (3) we conduct an empirical study in the Japanese stock market and prove the effectiveness of this method.

The rest of this paper is structured as follows. First, in Section 2 we introduce the volatility estimators that will be used in this study. Second, in Section 3, we present all the base models employed in this study and the methodology that will be used to combine the models. Next, in Section 4, we conduct an empirical analysis of our dataset and report and discuss the results. Finally, we present the conclusions in Section 5.

## 2. Volatility Estimators
### 2.1. Realized Volatility

We consider a log-price $p_t$ of a single asset, which follows a stochastic process as follows:

$$dp_t = \mu_t dt + \sigma_t dW_t, \tag{1}$$

where $\mu_t$ is the drift term, $\sigma_t$ is the instantaneous volatility term, and $W_t$ is a standard Brownian motion.

In this price process, the latent volatility is the integrated variance (IV), and the one-day IV is defined by the following equation:

$$IV_t = \int_{t-1}^{t} \sigma^2(\omega) d\omega. \tag{2}$$

In this equation, the $\sigma^2(\omega)$ is the instantaneous volatility at time $\omega$.

As mentioned above, daily IV is unobservable and daily RV computed from high-frequency return data is used as an estimator of IV (Andersen et al. 2001a, 2001b; Barndorff-Nielsen and Shephard 2002).

The definition of *RV* within a day is as follows:

$$RV_t^d = \sum_{i=1}^{M} r_{t,i}^2, \tag{3}$$

where $r_{t,i} = p_{t,i} - p_{t,i-1}$ is the return of the $i$th subinterval, $M$ is the number of subinterval on the day $t$, and $p_{t,i}$ is the logarithm of the price at time point $i$ in day $t$.

Therefore, the weekly and monthly average RVs are as follows

$$RV_t^w = \frac{1}{5}(RV_t^d + RV_{t-1}^d + \cdots + RV_{t-4}^d). \tag{4}$$

$$RV_t^m = \frac{1}{22}(RV_t^d + RV_{t-1}^d + \cdots + RV_{t-21}^d). \tag{5}$$

### 2.2. Realized Semivariance

According to Barndorff-Nielsen et al. (2008), the realized semi-volatility is primarily used to measure the positive and negative variation in the returns. The daily positive RV($RV^+$) is as follows:

$$RV_t^{d+} = \sum_{i=1}^{M} (\max(r_{t,i}, 0))^2, \tag{6}$$

and the daily negative RV($RV^-$) is as follows:

$$RV_t^{d-} = \sum_{i=1}^{M}(\min(r_{t,i}, 0))^2.$$

(7)

Furthermore, by definition, $RV$ can be decomposed into $RV^-$ and $RV^+$, as follows:

$$RV_t^d = RV_t^{d+} + RV_t^{d-}.$$

(8)

This decomposition holds for any given point in time.

### 2.3. Signed Jump

$SJ$ was introduced by Patton and Sheppard (2015), and the daily $SJ$ is defined by the following equation:

$$SJ_t^d = RV_t^{d+} - RV_t^{d-}.$$

(9)

When the positive price fluctuation is greater than the negative price fluctuation, $SJ$ is positive, and when the negative fluctuation is greater than the positive fluctuation, $SJ$ is negative.

## 3. Basic Models

### 3.1. Heterogeneous Autoregressive Model

The HAR-RV model is widely used in finance to predict the RV of financial assets using high-frequency data. It was proposed by Corsi (2009). It is a time-varying autoregressive model that uses lagged RV at different time scales as predictors to forecast RV. The most classical HAR-RV model is the HAR-RV (1, 5, 22) model, which uses three predictors—daily RV, weekly average RV, and monthly average RV.

$$RV_t^d = \beta_0 + \beta_d RV_{t-1}^d + \beta_w RV_{t-1}^w + \beta_m RV_{t-1}^m.$$

(10)

### 3.2. HAR-RSV and HAR-SJ Model

Based on realized semivariance (RSV), Patton and Sheppard (2015) have proposed a HAR-semivariance (HAR-SV) model. This model decomposes the daily $RV$ into $RV^+$ and $RV^-$ by Equation (8), and then uses the $RV^+$ and $RV^-$ to construct the HAR model. The model is defined as follows:

$$RV_t = \beta_0 + \beta_1^{d+} RV_{t-1}^{d+} + \beta_1^{d-} RV_{t-1}^{d-} + \beta_2 RV_{t-1}^w + \beta_3 RV_{t-1}^m$$

(11)

We include the HAR-RV-with-jumps (HAR-RV-J) model, which was proposed by Andersen et al. (2007), in our study. Then, we replace the jump component with the SJ component, because Patton and Sheppard (2015) found that a signed jumps model is helpful for forecasting volatility, to create a model we call the HAR-SJ model. The model is defined as follows:

$$RV_t = \beta_0 + \beta_1 RV_{t-1}^d + \beta_2 RV_{t-1}^w + \beta_3 RV_{t-1}^m + \beta_4 SJ_{t-1}^d$$

(12)

### 3.3. The HARQ Model

RV aims to estimate the IV of assets over a given period. However, the $M$ of Equation (3) has an upper bound, causing an estimation error. According to Barndorff-Nielsen and Shephard (2002), RV can be expressed as follows:

$$RV_t = IV_t + \eta_t, \eta_t \sim MN(0, \frac{2IQ_t}{M}),$$

(13)

where $MN$ denotes mixed normal, and $IQ_t$ is the integrated quarticity (IQ), which is defined as follows:

$$IQ_t = \int_{t-\Delta t}^{t} \sigma^4(\omega)d\omega. \tag{14}$$

Then, the IQ can be estimated by the RQ

$$RQ_t = \sum_{i=1}^{M} r_{t,i}^4. \tag{15}$$

Following Bollerslev et al. (2016), we can construct the HARQ model as follows:

$$RV_t = \beta_0 + (\beta_1 + \beta_{1Q}RQ_{t-1}^{1/2})RV_{t-1} + \beta_2 RV_{t-1}^w + \beta_3 RV_{t-1}^m. \tag{16}$$

We refer to HAR-RV, HAR-SV, HAR-SJ, and HARQ collectively as HAR-X models.

*3.4. Genetic Algorithms and Support Vector Regression*

SVR is an extension of the classification method SVM that was introduced in 1995 (Cortes and Vapnik 1995). SVR minimizes the error to obtain the regression equation by setting two parallel lines and enclosing the region between these two lines as tightly as possible around the output values. When data are difficult to fit in lower dimensions, one of the advantages of SVR is that it can map data in lower dimensional spaces to higher dimensional spaces by means of a kernel function, allowing the model to fit the data better. In the SVR training procedure, given a $T$ days dataset $F = \{(x_1, y_1), (x_2, y_2), \dots, (x_T, y_T)\}$, where $x_i$ is the training vector, $x_i \in R^n$, and $y_i$ is the output value, $y_i \in R$, the one-day ahead forecasting of SVR in time $t$ can be expressed as a linear equation as follows:

$$\hat{y}_{t+1} = \sum_{j=1}^{n} \omega_{t,j} x_{t,j} + b_t, \tag{17}$$

where $b_t \in R$, the predictor used in the HAR-X models will be used to obtain SVR models, so $n = 3$ or $4$, depending on the type of HAR-X model (only in the HAR-RV model is $n = 3$), $x_{t,j}$ is the $j$th predictor in time $t$, and $\omega_{t,j}$ is the coefficient of $x_{t,j}$.

To obtain the $\omega_{t,j}$, SVR transforms the regression problem into a convex optimization problem,

$$\min \frac{1}{2} \|\omega_t\|^2, \tag{18}$$

subject to

$$\begin{aligned}
\hat{y}_i - y_i &\leq \epsilon \\
y_i - \hat{y}_i &\leq \epsilon, \quad i = 1, 2, 3, \dots, t
\end{aligned} \tag{19}$$

where $\epsilon$ represents how large errors can be tolerated in regression tasks; $y_i$ is the actual value; and $\hat{y}_i$ is the predicted value on the day $i$. Moreover, to allow some errors, Bennett and Mangasarian (1992) introduced the soft margin method, which uses the slack variables $\xi_t$ and $\xi_t^*$ to relax the restriction in Equation (19), and Cortes and Vapnik (1995) used this in SVM. Then, the optimization problem is as follows:

$$\min \frac{1}{2} \|\omega_t\|^2 + C(\sum_{i=1}^{T} \xi_i + \xi_i^*), \tag{20}$$

subject to

$$\begin{aligned}
\hat{y}_i - y_i &\leq \epsilon + \xi_i^* \\
y_i - \hat{y}_i &\leq \epsilon + \xi_i \\
\xi_i, \xi_i^* &\geq 0, \quad i = 1, 2, 3, \dots, t
\end{aligned} \tag{21}$$

where the penalty coefficient $C > 0$ is a constant. Solving this optimization problem by constructing a Lagrange function, we can obtain the time $t$ coefficients $\omega_{t,i}$ (for details see Smola and Schölkopf 2004; Cortes and Vapnik 1995). Through the kernal function, SVR can be used to construct nonlinear time series problems. Furthermore, by choosing different kernel functions, SVR can construct different SVMs to obtain different regression equations. Commonly used kernel functions include the linear kernel function, polynomial kernel function, and radial basis function (RBF).

$$K(\boldsymbol{x_i}, \boldsymbol{x_j}) = \exp(-\gamma \|\boldsymbol{x_i} - \boldsymbol{x_j}\|^2), i, j = 1, 2, 3, \ldots, t. \tag{22}$$

Among them, the RBF kernel function (Equation (22)) is the most adaptable to various problems. Parameter $\gamma$ describes the variance of the kernel function, and together with the penalty coefficient $C$ and the sensitivity $\epsilon$, it affects the model's fitting ability to the data. Thus, the aim of optimizing SVR is to find the optimal values of these three parameters $\gamma, C$, and $\epsilon$[1].

GA is a search algorithm based on the principles of natural selection in biological evolution; Whitley (1994) provided a tutorial on the GA.

The basic principle of GA is to use a fitness function to evaluate individuals, eliminate those with low fitness, and retain those with the highest fitness to generate the next generation of individuals, thereby iteratively searching for the optimal solution. In our approach, we use the GA method to iteratively search for the optimal values of the three parameters $(\gamma, C, \epsilon)$ of the SVR to automatically optimize and find the best model.

*3.5. Hybrid Model*

In this study, we propose two types of hybrid HAR-X-SVR models, called HAR-X-SVR-1 and HAR-X-SVR-2, respectively. Following Pai and Lin (2005), our HAR-X-SVR-1 model decomposes the RV of returns into linear and nonlinear parts, as presented in the following equation:

$$RV_t^d = \hat{RV}_{t,L}^d + \hat{RV}_{t,N}^d + \epsilon_t, \tag{23}$$

where $\hat{RV}_{t,L}^d$ is the linear part prediction; $\hat{RV}_{t,N}^d$ is the nonlinear part prediction; and $\epsilon$ is the residual. Then, we use the HAR-X model to predict the linear part and assume that the HAR-X model can capture all the linear information. Therefore, let $f_{HAR,t}$ and $f_{SVR,t}$ represent the predictions of the HAR-X model and the SVR model for day $t$, respectively; then, the remaining part is the nonlinear part and the residual, as presented in the following equation:

$$RV_{t,N}^d = RV_t^d - f_{HAR,t} = f_{SVR,t} + \epsilon_t, \tag{24}$$

$$\hat{RV}_{t,N}^d = f_{SVR,t}. \tag{25}$$

In the HAR-X-SVR-2 model, we construct the HAR-X-SVR model by combining the HAR and SVR with a GA-optimized weight $\alpha$. The prediction of RV is calculated as below:

$$RV_t^d = \alpha f_{HAR,t} + (1 - \alpha) f_{SVR,t} + \epsilon_t. \tag{26}$$

In addition to searching for the three parameters of SVR using GA, we search for the weight $\alpha$ for the HAR-X-SVR-2 model (see Algorithm 1).

---

**Algorithm 1** HAR-X-SVR Model for *T* days forecasting

---

 1: **procedure** ONE HAR-X-SVR MODEL
 2:     Estimate the HAR model
 3:    **for** $t \leq T$ **do**
 4:      **for** $i \leq n$ **do**
 5:        **if** the model **is** HAR-X-SVR-1 **then**
 6:          Calculate residuals of HAR
 7:          Use residuals to search optimal $\epsilon,\gamma,C$ of SVR
 8:        **else if** the model **is** HAR-X-SVR-2 **then**
 9:          Search optimal $\epsilon,\gamma,C,\alpha$ for the HAR-X-SVR-2 model
10:        **end if**
11:        Forecast one-day ahead RV
12:        $i = i + 1$
13:      **end for**
14:      Calculate the mean value of *n* times forecasting as the forecasted value of day *t*
15:      $t = t + 1$
16:    **end for**
17:     Collect all *T* days forecasted values.
18: **end procedure**

---

## 4. Empirical Analysis

### 4.1. Data Description

In this study, our dataset was collected from the high-frequency intraday price data of the TOPIX and five individual stocks on TOPIX 30 from 1 January 2020 to 30 December 2022. We calculated the daily RV and other volatility estimators using the five-minute returns data. The computation method is presented in Equations (3), (6), (7), and (9). There are total 731 observations in our dataset.

In Table 1, we provide a summary of the data, including the maximum, minimum, and standard deviation of RV for TOPIX and individual stocks, as well as the 5%, 50%, and 95% quantiles of RV. Notably, the maximum and 95% quantile values of RV exhibit great variation because of the inclusion of the stock market crash in early 2020 due to the impact of the coronavirus pandemic.

**Table 1.** Data summary.

| Company | Mean | std | min | 5% | 50% | 95% | max |
|---------|------|-----|-----|-----|-----|-----|-----|
| TOPIX | 1.093 | 1.752 | 0.077 | 0.166 | 0.604 | 3.133 | 27.258 |
| 2914 | 0.644 | 0.909 | 0.070 | 0.162 | 0.405 | 1.798 | 13.090 |
| 8802 | 1.965 | 3.707 | 0.222 | 0.464 | 1.164 | 4.185 | 51.835 |
| 8411 | 1.107 | 1.731 | 0.129 | 0.233 | 0.699 | 3.050 | 23.436 |
| 8316 | 1.077 | 2.152 | 0.101 | 0.219 | 0.653 | 2.497 | 40.756 |
| 9432 | 1.904 | 4.043 | 0.134 | 0.353 | 0.955 | 5.696 | 56.830 |

Figures 1 and 2 depict the RV of our datasets, with the vertical coordinate being the RV and the horizontal coordinate being the date, where the blue and red lines represent the 95% quantile value and the average value, respectively. The graphs reveal that all datasets reached a high level at the beginning of the coronavirus pandemic, with long periods exceeding the 95% quantile value. Among them, 8802, 8316, and 8411 reached a high level at the end of 2022 again but not as high as at the beginning of the coronavirus pandemic, and the extreme RV value of 8316 occurred at the end of 2022. Furthermore, 8802, 8316, 8411, and TOPIX had a high level of RV at the end of the other periods, and they all had relatively stable levels of RV during the other periods, whereas 2914 and 9432 had a few higher peaks.

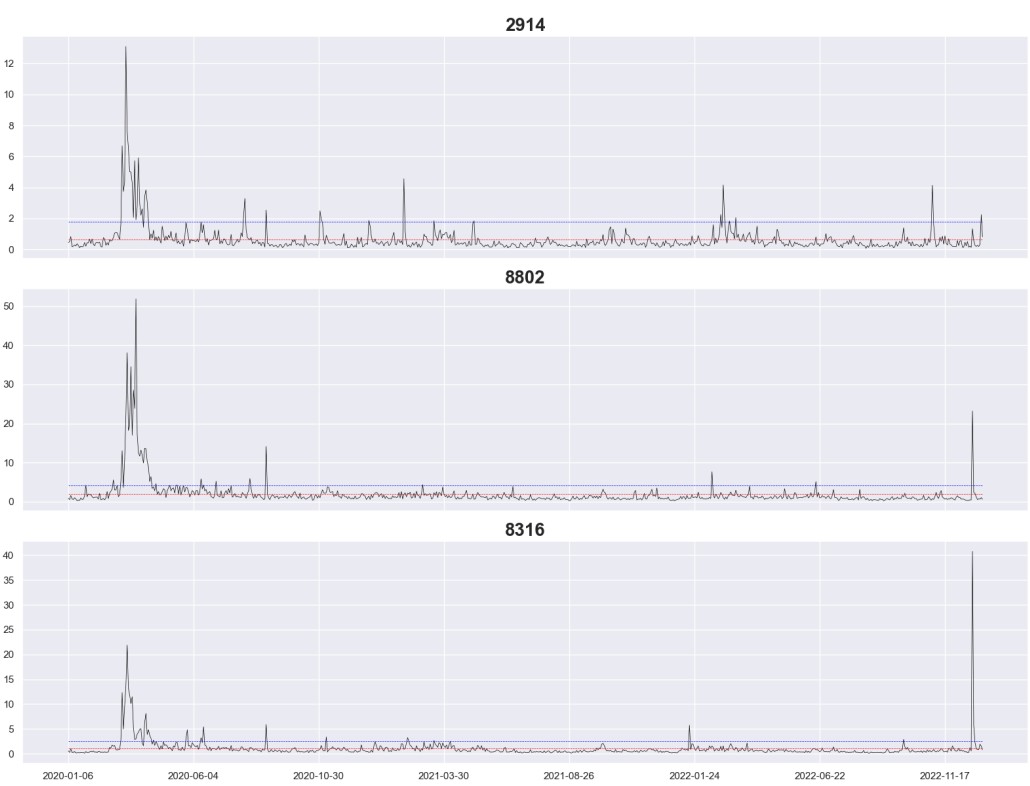

**Figure 1.** Realized volatility for stocks 2914, 8802, and 8316.

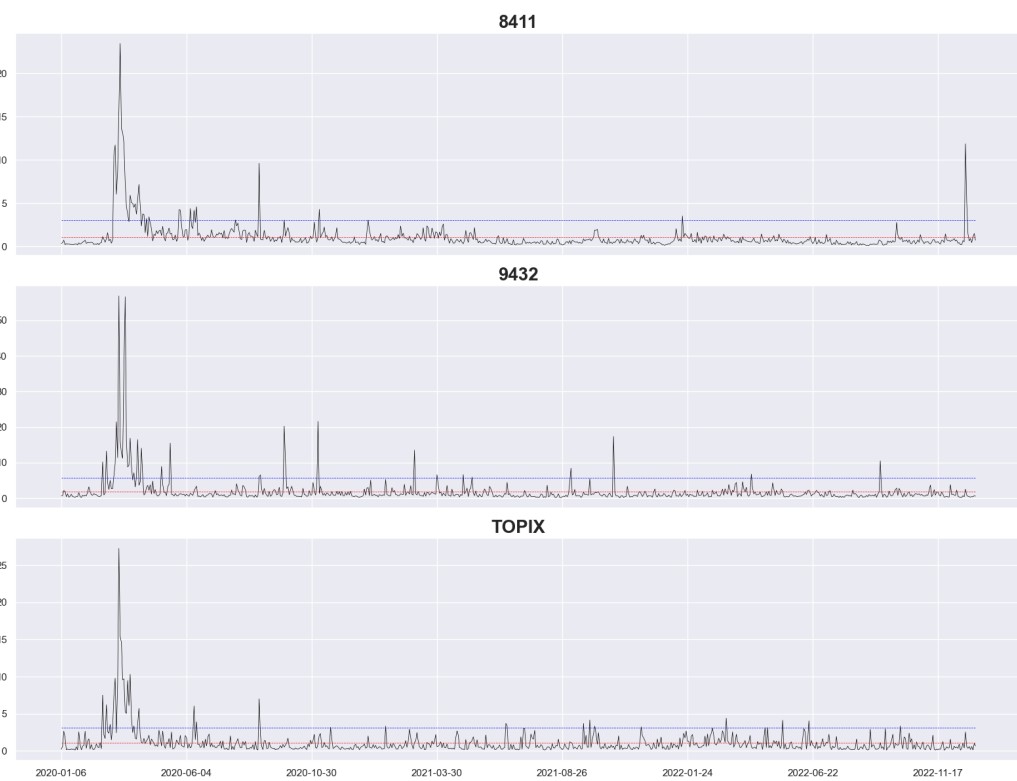

**Figure 2.** Realized volatility for stocks 8411 and 9432 and TOPIX as a whole.

*4.2. Out-of-Sample Forecasting*

In this study, we used the first 400 days data to train the models and the remaining 331 days data for the out-of-sample forecasting test. A fixed-length rolling window (RW) approach was used to update the different models each day, and we also used an increasing

window (IW) approach to train the models. The training sizes of the IW are from 400 to 730 days. Such a design is intended to consider the possible effects of different windowing approaches on the prediction results.

Throughout the training process, for HAR-X-SVR-1, we estimate HAR-X, derive the in-sample residuals, and use the in-sample residuals as the dependent variable to train the SVR. For HAR-X-SVR-2, we use the predictor and dependent variables that are identical to the corresponding HAR-X.

We employed four basic models: HAR-RV, HAR-SV, HAR-SJ, and HARQ. This resulted in 3 models under each basic model and 12 models in total. Among these models, the HAR-X model is estimated using OLS. After obtaining predictions from all the models, we calculated the loss function of the models in 22-day intervals as a sample for comparing the performance of each model with model confidence set (MCS). Patton (2011) finds that the mean square error (MSE) and the quasi-likelihood (Q-LIKE) are robust when comparing forecasting models. Following this paper, we utilized two loss functions—MSE and Q-LIKE—which are defined as follows:

$$
\begin{aligned}
MSE &= \frac{1}{n} \sum_{i=1}^{n} (\hat{y}_i - y_i)^2 \\
Q - LIKE &= \frac{1}{n} \sum_{i=1}^{n} (\log(\hat{y}_i) + \frac{y_i}{\hat{y}_i})
\end{aligned}
\tag{27}
$$

Model Confidence Set

The MCS method, introduced by Hansen et al. (2011), is an approach used to select the optimal predictive model. It addresses the challenge of comparing the performance of different models within different intervals, where reliance on a single loss function value alone may not be sufficient. The main procedure is to select a loss function for comparison and calculate the loss for all models in each period. In the case of time series models, a predetermined interval length is defined, and the loss is calculated for each interval within an extended period.

The set of all models is defined as $M_0$, and an additional set of models is denoted as $M(M \subset M_0)$. Then, an equivalence test $\delta$ and an elimination rule $e_M$ are defined. The $\delta$ is used to test whether the models in the set $M$ exhibit "good" performance based on the computed loss function. If any models in $M$ demonstrate poor performance, the $\delta$ hypothesis is rejected, and the elimination rule $e_M$ is applied to eliminate models. This process continues until the delta hypothesis is accepted.

Throughout the process, each model is assigned a *p*-value, with a *p*-value of 1 indicating the best model, and models that are eliminated earlier have lower *p*-values. For details on the specific steps, see the study by Hansen et al. (2011).

### *4.3. Results*

Based on different loss functions and windowing approaches, we report the *p*-values of our models for MCS on different datasets in Tables 2–5. The higher the *p*-value, the higher the performance of a model under the loss function—the highest is 1, and the lowest is 0. Moreover, we rank the models according to the *p*-values and sort the models according to the magnitude of the *p*-values.

In Table 2, we report the MCS *p*-values for the models we used the HAR-RV model to construct. The hybrid model significantly outperforms the HAR-RV model in datasets 2914, 8316, and 8411, and the HAR-RV model outperforms the hybrid model in dataset 2914 only under Q-LIKE when using the IW approach.

Furthermore, the hybrid model HAR-RV-SVR-1 significantly outperforms HAR-RV-SVR-2 only under MSE in dataset 8411 when using the RW approach. No hybrid model outperformed the HAR-RV model in the 8802 and 9432 datasets. In the TOPIX dataset, HAR-RV performs better under MSE, and HAR-RV-SVR-2 performs better under Q-LIKE. Using all datasets, we calculated the average rank of HAR-RV, HAR-RV-SVR1, and HAR-

RV-SVR-2; based on the average rank, HAR-RV-SVR-2 performs better than the other two models.

**Table 2.** MCS results for HAR-RV-based models.

| Company | Loss Function | RW | | | IW | | |
|---|---|---|---|---|---|---|---|
| | | **HAR-RV** | **HAR-RV-SVR-1** | **HAR-RV-SVR-2** | **HAR-RV** | **HAR-RV-SVR-1** | **HAR-RV-SVR-2** |
| **2914** | MSE | 0.012 | 0.012 | **1.000** | 0.656 | 0.656 | **1.000** |
| | Q-LIKE | 0.100 | 0.100 | **1.000** | **1.000** | 0.189 | 0.189 |
| | MSE rank | 2 | 2 | 1 | 2 | 2 | 1 |
| | Q-LIKE rank | 2 | 2 | 1 | 1 | 2 | 2 |
| | Average rank | 2 | 2 | 1 | 1.5 | 2 | 1.5 |
| **8316** | MSE | 0.037 | 0.037 | **1.000** | 0.230 | 0.230 | **1.000** |
| | Q-LIKE | 0.009 | 0.009 | **1.000** | 0.510 | 0.006 | **1.000** |
| | MSE rank | 2 | 2 | 1 | 2 | 2 | 1 |
| | Q-LIKE rank | 2 | 2 | 1 | 2 | 3 | 1 |
| | Average rank | 2 | 2 | 1 | 2 | 2.5 | 1 |
| **8411** | MSE | 0.540 | **1.000** | 0.540 | 0.671 | 0.512 | **1.000** |
| | Q-LIKE | 0.033 | 0.060 | **1.000** | 0.133 | 0.133 | **1.000** |
| | MSE rank | 2 | 1 | 2 | 2 | 3 | 1 |
| | Q-LIKE rank | 3 | 2 | 1 | 2 | 2 | 1 |
| | Average rank | 2.5 | 1.5 | 1.5 | 2 | 2.5 | 1 |
| **8802** | MSE | **1.000** | 0.105 | 0.279 | **1.000** | 0.055 | 0.687 |
| | Q-LIKE | **1.000** | 0.010 | 0.022 | **1.000** | 0.020 | 0.020 |
| | MSE rank | 1 | 3 | 2 | 1 | 3 | 2 |
| | Q-LIKE rank | 1 | 3 | 2 | 1 | 2 | 2 |
| | Average rank | 1 | 3 | 2 | 1 | 2.5 | 2 |
| **9432** | MSE | **1.000** | 0.047 | 0.047 | **1.000** | 0.037 | 0.037 |
| | Q-LIKE | **1.000** | 0.150 | 0.150 | **1.000** | 0.044 | 0.044 |
| | MSE rank | 1 | 2 | 2 | 1 | 2 | 2 |
| | Q-LIKE rank | 1 | 2 | 2 | 1 | 2 | 2 |
| | Average rank | 1 | 2 | 2 | 1 | 2 | 2 |
| **TOPIX** | MSE | **1.000** | 0.036 | 0.791 | **1.000** | 0.003 | 0.675 |
| | Q-LIKE | 0.103 | 0.103 | **1.000** | 0.054 | 0.054 | **1.000** |
| | MSE rank | 1 | 3 | 2 | 1 | 3 | 2 |
| | Q-LIKE rank | 2 | 2 | 1 | 2 | 2 | 1 |
| | Average rank | 1.5 | 2.5 | 1.5 | 1.5 | 2.5 | 1.5 |
| | Total average rank | 1.667 | 2.167 | 1.5 | 1.5 | 2.333 | 1.5 |

Note: The bold number is the best result among the three models under different loss function and window approach. The underlined number is the worst result among the three models.

The results for the HAR-SV-based models (Table 3) reveal that HAR-SV-SVR-2 significantly outperforms HAR-SV and HAR-SV-SVR-1, and HAR-SV only outperforms HAR-SV-SVR-2 under Q-LIKE in datasets 8802, 9432, and 2914 using the RW, IW, and IW approaches, respectively. After combining all rankings, we find that HAR-SV-SVR-2 significantly outperforms HAR-SV and HAR-SVR-1 in terms of average ranking.

**Table 3.** MCS result for HAR-SV-based models.

| Company | Loss Function | RW | | | IW | | |
|---|---|---|---|---|---|---|---|
| | | **HAR-SV** | **HAR-SV-SVR-1** | **HAR-SV-SVR-2** | **HAR-SV** | **HAR-SV-SVR-1** | **HAR-SV-SVR-2** |
| **2914** | MSE | 0.008 | 0.008 | **1.000** | 0.085 | 0.069 | **1.000** |
| | Q-LIKE | 0.010 | 0.010 | **1.000** | **1.000** | 0.496 | 0.496 |
| | MSE rank | 2 | 2 | 1 | 2 | 3 | 1 |
| | Q-LIKE rank | 2 | 2 | 1 | 1 | 2 | 2 |
| | Average rank | 2 | 2 | 1 | 1.5 | 2.5 | 1.5 |
| **8316** | MSE | 0.057 | 0.055 | **1.000** | 0.083 | 0.039 | **1.000** |
| | Q-LIKE | 0.245 | 0.245 | **1.000** | 0.085 | 0.085 | **1.000** |
| | MSE rank | 2 | 3 | 1 | 2 | 3 | 1 |
| | Q-LIKE rank | 2 | 2 | 1 | 2 | 2 | 1 |
| | Average rank | 2 | 2.5 | 1 | 2 | 2.5 | 1 |
| **8411** | MSE | 0.061 | 0.061 | **1.000** | 0.045 | 0.061 | **1.000** |
| | Q-LIKE | 0.000 | 0.014 | **1.000** | 0.104 | 0.104 | **1.000** |
| | MSE rank | 2 | 2 | 1 | 3 | 2 | 1 |
| | Q-LIKE rank | 3 | 2 | 1 | 2 | 2 | 1 |
| | Average rank | 2.5 | 2 | 1 | 2.5 | 2 | 1 |
| **8802** | MSE | 0.316 | 0.186 | **1.000** | 0.579 | 0.019 | **1.000** |
| | Q-LIKE | **1.000** | 0.417 | 0.417 | 0.295 | 0.089 | **1.000** |
| | MSE rank | 2 | 3 | 1 | 2 | 3 | 1 |
| | Q-LIKE rank | 1 | 2 | 2 | 2 | 3 | 1 |
| | Average rank | 1.5 | 2.5 | 1.5 | 2 | 3 | 1 |
| **9432** | MSE | 0.205 | 0.122 | **1.000** | 0.043 | 0.035 | **1.000** |
| | Q-LIKE | 0.396 | 0.103 | **1.000** | **1.000** | 0.798 | 0.798 |
| | MSE rank | 2 | 3 | 1 | 2 | 3 | 1 |
| | Q-LIKE rank | 2 | 3 | 1 | 1 | 2 | 2 |
| | Average rank | 2 | 3 | 1 | 1.5 | 2.5 | 1.5 |
| **TOPIX** | MSE | 0.088 | 0.011 | **1.000** | 0.060 | 0.007 | **1.000** |
| | Q-LIKE | 0.725 | 0.725 | **1.000** | 0.217 | 0.217 | **1.000** |
| | MSE rank | 2 | 3 | 1 | 2 | 3 | 1 |
| | Q-LIKE rank | 2 | 2 | 1 | 2 | 2 | 1 |
| | Average rank | 2 | 2.5 | 1 | 2 | 2.5 | 1 |
| | Total average rank | 2 | 2.417 | 1.083 | 1.917 | 2.5 | 1.167 |

Note: The bold number is the best result among the three models under different loss function and window approach. The underlined number is the worst result among the three models.

The results for the HAR-SJ-based models (Table 4) reveal that HAR-SJ-SVR-2 significantly outperforms HAR-SJ and HAR-SJ-SVR-1 in datasets 2914, 8316, and 8411; in contrast, in datasets 8802 and 9432, HAR-SJ slightly outperforms the other two hybrid models. In the TOPIX dataset, HAR-SJ performs better under MSE, and HAR-SJ-SVR-2 performs better under Q-LIKE. After combining all rankings, we find that HAR-SJ-SVR-2 outperforms the other two models.

**Table 4.** MCS result for HAR-SJ-based models.

| Company | Loss Function | RW | | | IW | | |
|---|---|---|---|---|---|---|---|
| | | **HAR-SJ** | **HAR-SJ-SVR-1** | **HAR-SJ-SVR-2** | **HAR-SJ** | **HAR-SJ-SVR-1** | **HAR-SJ-SVR-2** |
| **2914** | MSE | 0.003 | 0.003 | **1.000** | 0.274 | 0.274 | **1.000** |
| | Q-LIKE | 0.021 | 0.000 | **1.000** | **1.000** | 0.266 | 0.266 |
| | MSE rank | 2 | 2 | 1 | 2 | 2 | 1 |
| | Q-LIKE rank | 2 | 3 | 1 | 1 | 2 | 2 |
| | Average rank | 2 | 2.5 | 1 | 1.5 | 2 | 1.5 |
| **8316** | MSE | 0.056 | 0.049 | **1.000** | 0.171 | 0.075 | **1.000** |
| | Q-LIKE | 0.328 | 0.137 | **1.000** | 0.645 | 0.000 | **1.000** |
| | MSE rank | 2 | 3 | 1 | 2 | 3 | 1 |
| | Q-LIKE rank | 2 | 3 | 1 | 2 | 3 | 1 |
| | Average rank | 2 | 3 | 1 | 2 | 3 | 1 |
| **8411** | MSE | 0.048 | 0.333 | **1.000** | 0.363 | 0.363 | **1.000** |
| | Q-LIKE | 0.013 | 0.034 | **1.000** | 0.144 | 0.144 | **1.000** |
| | MSE rank | 3 | 2 | 1 | 2 | 2 | 2 |
| | Q-LIKE rank | 3 | 2 | 1 | 2 | 2 | 1 |
| | Average rank | 3 | 2 | 1 | 2 | 2 | 1 |
| **8802** | MSE | **1.000** | 0.067 | 0.067 | 0.686 | 0.129 | **1.000** |
| | Q-LIKE | **1.000** | 0.087 | 0.087 | **1.000** | 0.414 | 0.414 |
| | MSE rank | 1 | 2 | 2 | 2 | 3 | 1 |
| | Q-LIKE rank | 1 | 2 | 2 | 1 | 2 | 2 |
| | Average rank | 1 | 2 | 2 | 1.5 | 2.5 | 1.5 |
| **9432** | MSE | 0.033 | 0.011 | **1.000** | **1.000** | 0.001 | 0.145 |
| | Q-LIKE | 0.508 | 0.508 | **1.000** | **1.000** | 0.099 | 0.907 |
| | MSE rank | 2 | 3 | 1 | 1 | 3 | 2 |
| | Q-LIKE rank | 2 | 2 | 1 | 1 | 3 | 2 |
| | Average rank | 2 | 2.5 | 1 | 1 | 3 | 2 |
| **TOPIX** | MSE | **1.000** | 0.001 | 0.476 | **1.000** | 0.007 | 0.337 |
| | Q-LIKE | 0.331 | 0.151 | **1.000** | 0.472 | 0.066 | **1.000** |
| | MSE rank | 1 | 3 | 2 | 1 | 3 | 2 |
| | Q-LIKE rank | 2 | 3 | 1 | 2 | 3 | 1 |
| | Average rank | 1.5 | 3 | 1.5 | 1.5 | 3 | 1.5 |
| | Total average rank | 1.917 | 2.5 | 1.25 | 1.583 | 2.583 | 1.417 |

Note: The bold number is the best result among the three models under different loss function and window approach. The underlined number is the worst result among the three models.

Similar to the results for the HAR-SV-based models, the results for the HARQ-based models (Table 5) reveal that HARQ-SVR-2 outperforms the other two models, and the HARQ model outperforms the other two models in datasets 9432 and 2914 under Q-LIKE. However, in terms of average ranking, HARQ-SVR-2 significantly outperforms the other two models. After combining all four HAR-X-based models, we find that the average rankings of HAR-X, HAR-X-SVR-1, and HAR-X-SVR-2 are 1.896, 2.333, and 1.229 under the RW approach and 1.771, 2.375, and 1.375 under the IW approach, respectively. In both the RW and IW approaches, the HAR-X-SVR-2 significantly outperforms the other two types of models.

**Table 5.** MCS result for HARQ-based models.

| Company | Loss Function | RW | | | IW | | |
|---|---|---|---|---|---|---|---|
| | | **HARQ** | **HARQ-SVR-1** | **HARQ-SVR-2** | **HARQ** | **HARQ-SVR-1** | **HARQ-SVR-2** |
| **2914** | MSE | 0.031 | 0.031 | **1.000** | 0.560 | **1.000** | 0.977 |
| | Q-LIKE | 0.042 | 0.042 | **1.000** | **1.000** | 0.601 | 0.202 |
| | MSE rank | 2 | 2 | 1 | 3 | 1 | 2 |
| | Q-LIKE rank | 2 | 2 | 1 | 1 | 2 | 3 |
| | Average rank | 2 | 2 | 1 | 2 | 1.5 | 2.5 |
| **8316** | MSE | 0.064 | 0.064 | **1.000** | 0.065 | 0.065 | **1.000** |
| | Q-LIKE | 0.114 | 0.070 | **1.000** | 0.515 | 0.034 | **1.000** |
| | MSE rank | 2 | 2 | 1 | 2 | 2 | 1 |
| | Q-LIKE rank | 2 | 3 | 1 | 2 | 3 | 1 |
| | Average rank | 2 | 2.5 | 1 | 2 | 2.5 | 1 |
| **8411** | MSE | 0.039 | 0.039 | **1.000** | 0.051 | 0.054 | **1.000** |
| | Q-LIKE | 0.000 | 0.000 | **1.000** | 0.019 | 0.019 | **1.000** |
| | MSE rank | 2 | 2 | 1 | 3 | 2 | 1 |
| | Q-LIKE rank | 2 | 2 | 1 | 2 | 2 | 1 |
| | Average rank | 2 | 2 | 1 | 2.5 | 2 | 1 |
| **8802** | MSE | 0.048 | 0.048 | **1.000** | 0.050 | 0.050 | **1.000** |
| | Q-LIKE | 0.143 | 0.140 | **1.000** | 0.100 | 0.100 | **1.000** |
| | MSE rank | 2 | 2 | 1 | 2 | 2 | 1 |
| | Q-LIKE rank | 2 | 3 | 1 | 2 | 2 | 1 |
| | Average rank | 2 | 2.5 | 1 | 2 | 2 | 1 |
| **9432** | MSE | 0.225 | 0.130 | **1.000** | 0.569 | 0.512 | **1.000** |
| | Q-LIKE | **1.000** | 0.276 | 0.276 | **1.000** | 0.622 | 0.273 |
| | MSE rank | 2 | 3 | 1 | 2 | 3 | 1 |
| | Q-LIKE rank | 1 | 2 | 2 | 1 | 2 | 3 |
| | Average rank | 1.5 | 2.5 | 1.5 | 1.5 | 2.5 | 2 |
| **TOPIX** | MSE | 0.232 | 0.361 | **1.000** | 0.157 | 0.300 | **1.000** |
| | Q-LIKE | 0.153 | 0.153 | **1.000** | 0.062 | 0.062 | **1.000** |
| | MSE rank | 3 | 2 | 1 | 3 | 2 | 1 |
| | Q-LIKE rank | 2 | 2 | 1 | 2 | 2 | 1 |
| | Average rank | 2.5 | 2 | 1 | 2.5 | 2 | 1 |
| | Total average rank | 2 | 2.25 | 1.083 | 2.083 | 2.083 | 1.417 |

Note: The bold number is the best result among the three models under different loss function and window approach. The underlined number is the worst result among the three models.

*4.4. Discussion*

4.4.1. Summary

Regarding return volatility forecasting, in addition to the classic (G)ARCH and ARMA models, there has been an influx of research on forecasting using various types of machine learning models with the development of machine learning technology. Sezer et al. (2020) provided an exhaustive introduction to the application of deep learning models in financial time series forecasting. In recent years, some studies have attempted to combine statistical and machine learning models to improve forecasting accuracy (e.g., Zhang and Qiao 2021; Kim and Won 2018). However, they still take a human-set approach to selecting hyperparameters for machine learning models, which is time-consuming and labor-intensive and requires much experience in adjusting the parameters to the optimal level. Waring et al. (2020) provided an exhaustive overview of the application of machine learning and automated machine learning to time series forecasting. They also compared different

frameworks and techniques, although they did not conduct an empirical analysis. Therefore, in this study, we propose the use of automatic machine learning methods to solve the problem of setting hyperparameters when using machine learning, thus simplifying the parameter tuning of machine learning. Our model is not limited to hyperparameter optimization but also includes the selection of weights when combining the forecasts from the HAR and SVR models with an automatic optimization algorithm. The weights of the model are dynamic and are updated daily according to the training set.

4.4.2. Empirical Results

We compare the two hybrid models with the regular HAR-X model in the Japanese stock market; Kim and Won (2018) and Zhang and Qiao (2021) analyzed the performance of the hybrid model in the Korean and Chinese stock markets, respectively. Their studies revealed that the hybrid model is helpful for improving RV forecasting accuracy. In our empirical study, we use six datasets, including an index dataset (TOPIX) and five individual stock datasets (stock symbol: 2914, 8316, 8411, 8802, 9432). Among these six datasets, we find that our HAR-X-SVR-2 model is consistently the best-performing model when using the index dataset (TOPIX) and outperforms the other two models in 12 out of 16 cases. In the individual stock dataset, it outperforms the other models in 66 out of 80 cases and 78 out of a total of 96 cases , which is 81.25% of the total[2]. Using different window approaches, no significant effect is observed on the ranking of the models, with HAR-X-SVR-2 outperforming the other two models in 37 out of 48 cases for RW and 41 out of 48 cases for IW. Among the three results where the base model is used with more information (HAR-SV, HARQ, and HAR-SJ), the average ranking of the HAR-X-SVR-2 model is higher than the average ranking when simply using the HAR-RV model as the base model, i.e., when the model has more information, the HAR-X-SVR2 model tends to better utilize the information contained in the predictor than the other two models. Although the other hybrid model has been used by Pai and Lin (2005), which had good performance in stock price prediction, it did not outperform the HAR model in RV prediction. Therefore, at least in the Japanese stock market, the second type of hybrid model performs more reliably than the others. Future research can study a greater number of models to expand the predicted values to more combinations of models while considering more automatic machine learning frameworks.

4.4.3. Limitations and Future Research

The limitations of this study are as follows. This study did not empirically analyze the model's reliability in other countries, such as the UK and US, and other financial markets, such as FX markets. Additionally, the correlation between assets was not considered in this study. Moreover, this study does not explore certain other machine learning or automated machine learning models.

To address the limitations, we make several suggestions for future research. First, more empirical studies should be conducted in different countries and markets for different financial products to confirm whether the hybrid model proposed in this study is generalizable to more markets. Second, multivariate models should be employed to consider correlations across markets for different assets. Finally, more machine learning models and automated machine learning frameworks should be applied. This will enable researchers to provide a more detailed analysis of the effectiveness of hybrid models in the future.

**5. Conclusions**

This study proposed the use of HAR-X-SVR models, which are used for RV prediction, and tested their out-of-sample forecasting performance under TOPIX and five individual stocks datasets. It constructed two types of hybrid models by combining four basic HAR-X models with SVR using two different combining methods, and these two types of hybrid models were compared with basic HAR-X models. In the Japanese stock market, the empirical results revealed that although the first hybrid model is effective in improving

model accuracy in the stock price prediction study by Pai and Lin (2005), the first hybrid model is not significantly effective in improving model accuracy in forecasting RV. However, our second hybrid model performs very well in all six datasets, especially when based on the HAR-SV and HARQ models. Based on the empirical results, we suggest that combining different machine learning models with time series models can be useful for improving prediction accuracy in the Japanese stock market. Nevertheless, confirming the feasibility and viability of such an approach in different markets requires a more comprehensive and meticulous investigation.

We have proposed the following recommendations for future research in order to more comprehensively examine the hybrid model's credibility. In this study, we only considered hybrid models to enhance the forecasting ability of HAR models. Other volatility models, such as MEM models (Engle and Gallo 2006), can be considered in subsequent work. Furthermore, automatic machine learning methods have developed rapidly in recent years, so the application of machine learning frameworks such as AutoGluon (Erickson et al. 2020) can be considered in subsequent studies. Our approach can be generalized to multivariate analysis using multivariate models such as the multivariate-HARQ (M-HARQ) model (Bollerslev et al. 2018) for multivariate forecasting while considering correlations between different markets.

**Author Contributions:** Conceptualization, Y.Z. and T.M.; software, Y.Z.; data curation, Y.Z. and T.M.; formal analysis, Y.Z.; writing—original draft preparation, Y.Z.; writing—review and editing, Y.Z. and T.M.; supervision, T.M. All authors have read and agreed to the published version of the manuscript.

**Funding:** This study is partly supported by the Institute of Statistical Mathematics (ISM) cooperative research program [2023-ISMCRP-2034] and JSPS KAKENHI [Grant Number 21K01433].

**Data Availability Statement:** Restrictions apply to the availability of these data. Data was obtained from Nikkei Media Marketing, Inc.

**Conflicts of Interest:** The authors declare no conflicts of interest.

## Notes

[1]  For more introduction to property of RBF, see Micchelli (1986) and Boser et al. (1992)
[2]  The results obtained from different datasets with different loss functions, basic model and windowing approach are considered as a case, e.g., in the case of the results with HAR-RV as the basic model, the results of MSE obtained with the TOPIX dataset and under the RW approach are a case.

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
