# Peer review of "A Hybrid Model for Forecasting Realized Volatility Based on Heterogeneous Autoregressive Model and Support Vector Regression"

_risks, doi:10.3390/risks12010012_

Round 1

Reviewer 1 Report

Comments and Suggestions for Authors

A Hybrid Model for Forecasting Realized Volatility based on Heterogeneous Autoregressive Model and Support Vector Regression

risks-2790519

The authors investigate two types of hybrid models based on the heterogeneous autoregressive (HAR) model and support vector regression (SVR) model to forecast realized volatility (RV).

My specific and suggestions comments are listed below.

1-     The exposition of the paper should be entirely improved. The authors should put more emphasize on the utility of this scientific contribution because I find that authors mainly focus on general aspects of the econometric analysis, which might not be enough to make general conclusions. Moreover, there is a need to reveal the uniqueness of their research question and the WHY question. That is, what makes their paper so unique and so implications-oriented that one should read it.

2-     There is a lot of anecdotal evidence without scientific facts in the paper (For example, line 53, 81,

3-     Explain the second type of hybrid model and how the predicted value is considered in the equation!!

4-     What’s the GA method? Very high number of acronyms are used in this paper that can get lost while reading. I believe that acronyms should be used far less in this paper.

Reviewer 2 Report

Comments and Suggestions for Authors

Many stock return volatility models have been developed, including this study, which proposes two types of hybrid models: the HAR and SVR.

• A broader and in-depth literature review of stock return volatility models still needs to be continued to strengthen the originality, novelty, and contribution of this study

• Additional explanation is needed to see the common thread of the relationship between the models developed with GARCH and realized volatility and ARMA models

• The forecasting results of the developed hybrid model need to be discussed further, showing the strengths and weaknesses compared to previous models that have been developed previously.

Reviewer 3 Report

Comments and Suggestions for Authors

Comments on the Quality of English Language

The English language needs little editing.

Round 2

Reviewer 1 Report

Comments and Suggestions for Authors

The revised version of this paper has improved significantly. Authors have adressed all my concerns and comments. Therefore, I recommend accepting the paper